

# A three-stage heuristic task scheduling for optimizing the service level agreement satisfaction in device-edge-cloud cooperative computing

Yongxuan Sang[1], Junqiang Cheng[2], Bo Wang[1] and Ming Chen[1]

[1] Zhengzhou University of Light Industry, Zhengzhou, China
[2] Europe-Aisa Hi-tech and Digital Technology Company Limited, Zhengzhou, China

## ABSTRACT

Device-edge-cloud cooperative computing is increasingly popular as it can effectively address the problem of the resource scarcity of user devices. It is one of the most challenging issues to improve the resource efficiency by task scheduling in such computing environments. Existing works used limited resources of devices and edge servers in preference, which can lead to not full use of the abundance of cloud resources. This article studies the task scheduling problem to optimize the service level agreement satisfaction in terms of the number of tasks whose hard-deadlines are met for device-edge-cloud cooperative computing. This article first formulates the problem into a binary nonlinear programming, and then proposes a heuristic scheduling method with three stages to solve the problem in polynomial time. The first stage is trying to fully exploit the abundant cloud resources, by pre-scheduling user tasks in the resource priority order of clouds, edge servers, and local devices. In the second stage, the proposed heuristic method reschedules some tasks from edges to devices, to provide more available shared edge resources for other tasks cannot be completed locally, and schedules these tasks to edge servers. At the last stage, our method reschedules as many tasks as possible from clouds to edges or devices, to improve the resource cost. Experiment results show that our method has up to 59% better performance in service level agreement satisfaction without decreasing the resource efficiency, compared with eight of classical methods and state-of-the-art methods.

# INTRODUCTION

With the development of computer and communications technology as well as the growing need for the quality of human life, smart devices, *e.g.*, smartphones and Internet of Thing (IoT) devices, have become more and more popular. As shown in the Cisco Annual Internet Report (CAIR) (*Cisco, 2020*) released in March 2020, networked devices will be increased from 18.4 billion in 2018 to 29.3 billion in 2023, and IoT devices will account for 50 percent by 2023. Juniper Research has reported similar results in 2018, IoT devices will grow at 140% over the next 4 years (*Sorrel, 2018*). Because of limited resources, plenty of

Corresponding author
Bo Wang, wangb@zzuli.edu.cn

user devices cannot satisfy their respective requirements at most time (*Ghasempour, 2019*; *Liu et al., 2019b*), as Internet applications has undergone rapid growth in both variety and complexity with the development of artificial intelligence algorithms (*e.g.*, deep neural networks) and communication technologies (*e.g.*, 5G and Wifi 6) (*Wang et al., 2020b*).

To address the resource scarcity problem of user devices, several researchers exploited the low latency edge resources (*Balasubramanian et al., 2020*) and the abundant cloud resources (*Strumberger et al., 2019*). Only one of them cannot address the problem effectively due to either the limited resource of edges or the poor network performance of clouds (*Wang et al., 2019b*). Thus, with the integration of respective benefits of edge and cloud computing, device-edge-cloud cooperative computing (DE3C) (*Hong et al., 2019*) is an effective way, where edges and clouds are employed jointly for expanding the resource capacity of user devices.

Task scheduling or offloading is an effective way for optimizing the task performance and the resource efficiency for DE3C, which decides the location (the corresponding device, an edge or a cloud) where each task to be processed (offloading decision) and the computing resources which each task performs on in a specified order (task assignment and ordering) (*Wang et al., 2020b*; *Islam et al., 2021*). Therefore, several works have proposed various task scheduling methods trying to optimize the response time (*Han et al., 2019*; *Meng et al., 2019*; *Meng et al., 2020*; *Apat et al., 2019*; *Ren et al., 2019*; *Liu et al., 2019a*; *Wang et al., 2021*), the resource cost (*Mahmud et al., 2020*; *Gao et al., 2019*; *Chen et al., 2019*) or the profit (*Chen et al., 2020*; *Yuan & Zhou, in press*) for providing services in DE3C. These works were concerned on addressing only one or two sub-problems of task scheduling, *e.g.*, offloading decision or/and task assignment, and thus cannot provide global optimal solutions. In addition, a lot of existed works did not employ local device resources without network latency, even though a lot of smart devices have been equipped with computing resources almost equivalent to personal computers (*Wu et al., 2019*), nowadays.

## Motivation

This paper focuses on the joint problem of offloading decision, task assignment and task ordering, to optimize the profit for service providers in DE3C, by improving the Service Level Agreement (SLA) satisfaction. An SLA is enforced when a user uses a provider's service. If the SLA is fail to be satisfied, the provider must pay a penalty, which reduces the provider's profit. In addition, an SLA violation reduces the provider's reputation, and thus may lead to a loss of some potential users (*Papadakis-Vlachopapadopoulos et al., 2019*), while plenty of related works only concerned on improving the response time or the resource cost, which is in contradiction to the SLA satisfaction optimization, as better response times or less resources generally result in a less number of completed tasks. In addition, to our best knowledge, all existed work prioritised local resources (devices) for processing tasks and rented resources from clouds only when local and edge resources are not enough, as local resources are cheap and have no network latency. They did not consider that some tasks processed locally or in edges can be assigned to clouds to save some local or edge resources for completing tasks that cannot be finished by cloud resources.

**Table 1  The information of DE3C system for the case motivating our work.**

**a. task requirements**

| Task | Computing resource amount | Transferred data amount | Deadline |
|------|---------------------------|-------------------------|----------|
| $t_1$ | 100 GHz | 100 Mbit | 50 s |
| $t_2$ | 100 GHz | 100 Mbit | 50 s |
| $t_3$ | 10 GHz | 100 Mbit | 10 s |
| $t_4$ | 10 GHz | 100 Mbit | 10 s |

**b. resource configurations**

| Resource | Computing capacity | Transmission bandwidth |
|----------|--------------------|------------------------|
| Device | 2 GHz | $\infty$ |
| Edge | 2 GHz | 100 Mbps |
| Cloud | 4 GHz | 10 Mbps |

For example, there are four tasks, $t_1$, $t_2$, $t_3$, and $t_4$, to be scheduled in a DE3C with one device, one edge server, and a cloud. The information of tasks and resources are shown in Table 1. The times consumed by $t_1$ and $t_2$ are 50 s, 51 s, and 35 s, respectively, when they are scheduled to the device, the edge server, and the cloud, and the consumed times are 5 s, 6 s, and 12.5 s, respectively, for $t_3$ and $t_4$. With the scheduling order of $t_1$, $t_2$, $t_3$, and $t_4$, and the idea of using device resources first, $t_1$, $t_3$, and $t_2$ are respectively scheduled to the device, the edge server, and the cloud, but the requirements of $t_4$ cannot be satisfied. But if scheduling tasks with the priority order of the cloud, the edge server, and the device, requirements of all tasks can be satisfied, where $t_1$ and $t_2$ are scheduled to the cloud and $t_3$ and $t_4$ are scheduled to the device.

## Contribution

This paper focuses on maximizing the SLA satisfaction, *i.e.,* optimizing the number of tasks whose hard-deadlines are met, in DE3C by task scheduling, by exploiting both the low network latency of edges and the rich computing resources of clouds. To address the task scheduling problem in DE3C, the paper formulates it into a binary nonlinear programming (BNLP) for the SLA satisfaction optimization. In order to solve the problem in polynomial time, a heuristic method is designed based on the idea of least accumulated slack time first (LASTF) (*Wang et al., 2020a*) and earliest deadline first (EDF) (*Benoit, Elghazi & Robert, 2021*). In brief, the contributions of this paper can be summarized as followings.

- The task scheduling problem of DE3C is formulated as a BNLP with two objectives, where the major one is maximizing the number of tasks whose requirements are satisfied[1] and the second one is maximizing the resource utilization.
- A three-stage heuristic task scheduling method (TSSLA) is designed for DE3C. The first stage is to exploit the abundant computing resource of clouds, to finish as many tasks as possible,[2] by pre-scheduling tasks in the resource priority order of clouds, edges, and devices. In the second stage, TSSLA reschedules tasks from edges to respective devices to make some edge resources free for completing more tasks, and schedules remaining unscheduled tasks to edges. At the last stage, our method reschedules tasks from clouds

[1]This paper considers the number of finished tasks with hard-deadlines as the SLA satisfaction metric. The proposed approach is compatible to any other metric.

[2]In this paper, finishing/completing a task means finishing the task within its deadline, *i.e.,* satisfying the task's SLA requirement.

to respective devices or edges, to reduce the resource cost. For task scheduling in each location, the proposed method respectively employs LASTF and EDF to assign the computing core for each task and decide the task execution order.

- Simulated experiments are conducted referring to recent related works and the reality, to evaluate our proposed heuristic method. Experiment results verify our method having a much better performance than eight of classical and state-of-the-art methods in optimizing SLA satisfaction.

The rest of this paper is organized as follows. The second section illustrates the related works. Third section present the formulation of the task scheduling problem this paper concerned. Fourth section presents the three-stage heuristic method designed, and analyses its time complexity. The subsequent section evaluates the performance of our task scheduling method by simulated experiments. And finally, the last section concludes this paper.

## RELATED WORK

As DE3C is one of the most effective ways to solve the problem of insufficient resources of smart devices and task scheduling is a promising technology to improve the resource efficiency, several researchers have focused on the design of efficient task scheduling methods in various DE3C environments (*Wang et al., 2020b*).

To improve the response time of tasks, the method proposed by *Apat et al. (2019)* iteratively assigned the task with the least slack time to the edge server closest to the user. Tasks are assigned to the cloud when they cannot be finished by edges. Their work did not consider the task scheduling on each server. OnDisc, proposed by *Han et al. (2019)*, heuristically dispatched a task to the server providing the shortest additional total weighted response time (WRT), and sees the cloud as a server, to improve overall WRT. *Stavrinides & Karatza (2019)* proposed a heuristic method for improving the deadline miss Ratio. Their proposed method respectively employed EDF and earliest finish time first for task selection and resource allocation, and tried to fill a task before the input data is ready for the next task to execute.

The above research focused on the performance optimization for task execution, but did not concern the cost of used resources. In general, a task requires more resources for a better performance, and thus there is a trade-off between the task performance and the resource cost. Therefore, several works concerned the optimization of the resource cost or the profit for service providers. For example, *Chen et al. (2020)* presented a task scheduling method to optimize the profit, where the value of a task was proportional to the resource amounts and the time it took, and resources were provided in the form of VM. Their proposed method first classified tasks based on the amount of its required resources by K-means. Then, their method used Kuhn–Munkres method to solve the optimal matching of tasks and VMs with profit maximization for the VM class and the task class closest to the VM class, where all VMs were seen as one VM class. This work ignored the heterogeneity between edge and cloud resources, which may lead to resource inefficiency (*Kumar et al., 2019*). *Li, Wang & Luo (2020)* tried to optimize the finish time

and the cloud resource usage cost. Their proposed method first made the offloading decision for each task, adopting the artificial fish swarm algorithm improved by simulated annealing method for calculating the probability of updating bulletin, to avoid falling into local optimal solution. Then their method greedily assigned a task to the computing node with the minimum utilization in edges or the cloud. This work focused on media delivery applications, and thus considered that each task can be divided into multiple same-sized subtasks for parallel process, which limited its application scope. The method proposed by *Mahmud et al. (2020)* iteratively assigned the offloaded application to the first computational instance such that all requirements are satisfied and the profit merit is minimum, where cloud-based instances were sorted behind edge-based instances and the profit merit was defined as the ratio of the profit and the slack time.

All of the aforementioned methods employed only edge and cloud resources for task processing, even though most of user devices have been equipped with various computing resources (*Wu et al., 2019*) which have zero transmission latency for users' data. To exploit all the advantages of the local, edge and cloud resources, some works are proposed to address the task scheduling problem for DE3C. The method presented in *Lakhan & Li (2019)* first tried several existed task order method, *e.g.*, EDF, EFTF, and LSTF, and selected the result with the best performance for task order. Then, the method used existed pair-wise decision methods, TOPSIS (*Liang & Xu, 2017*) and AHP (*Saaty, 2008*), to decide the position for each task's execution, and applied a local search method exploiting random searching for the edge/cloud. For improving the delay, the approach presented in *Miao et al. (2020)* first decided the amounts of data that is to be processed by the device and an edge/cloud computing node, assuming each task can be divided into two subtasks with any data size. Then they considered to migrate some subtasks between computing nodes to further improve the delay, for each task. The method proposed in *Zhang et al. (2019)* iteratively assigned the task required minimal resources to the nearest edge server that can satisfy all of its requirements. *Ma et al. (2022)* proposed a load balance method for improving the revenue for edge computing. The proposed method allocated the computing resources of the edge node with the most available cores and the smallest move-up energy to the new arrived task. To improve the total energy consumption for executing deep neural networks in DE3C with deadline constraints, *Chen et al. (2022)* proposed a particle swarm optimization algorithm using mutation and crossover operators for population update. *Wang et al. (2021)* leveraged reinforcement learning with sequence-to sequence neural network for improving the latency and the device energy in DE3C. Machine learning-based or metaheuristic-based approaches may achieve a better performance than heuristics, but in general, they consume hundreds to tens of thousands more time, which makes them not applicable to make online scheduling decisions.

All of these existed research concerned only one or two problems of offloading decision, task assignment, and task ordering, which leads to suboptimal solutions. In addition, they were not fully explored the advantage of abundant cloud resources, as they considered assigning tasks to the cloud only when local and edge resources are exhausted. To address these issues, this paper designs a heuristic method for optimizing SLA satisfaction in DE3C.

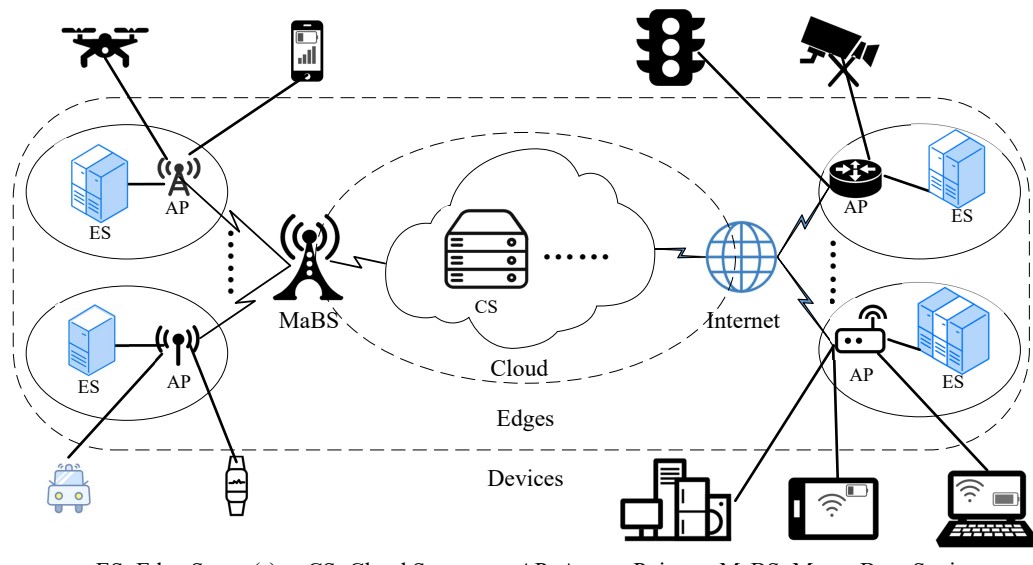

ES: Edge Server(s)    CS: Cloud Servers    AP: Access Point    MaBS: Macro Base Station

**Figure 1** **The DE3C environment considered in this paper.**

To our best knowledge, this is the first attempt to jointly address the problems of offloading decision, task assignment, and task ordering.

## PROBLEM FORMULATION

The DE3C environment considered in this paper is composed of various user devices, multiple edges (short for edge computing centers), and one cloud,[3] as shown in Fig. 1. Each device launches one or more tasks for processing data it collected from user behaviors or surroundings. Each task can be processed locally, or offloaded to an edge covering the device or the cloud. When a task is offloaded, its processed data must be transmitted from the device to the edge or the cloud before its processing, by various communication links, *e.g.*, wireless networks, telecommunications, etc. There have been some research on data transmission in advance to improve the network latency by predicting task offloading decisions (*Zhang et al., 2017*), which is complementary to our work. In this paper, the data is assumed to be transmitted only after the offloading decision is made for each task, which can avoid the waste of network resources due to the failure of predicting.

The considered DE3C system is consisted of $M$ user devices ($\mathcal{M} = \{m_1, m_2, .., m_M\}$), $E$ edges ($\mathcal{E} = \{e_1, e_2, \ldots, e_E\}$), and one public cloud. For device $m_j$, there are $n_j$ cores ($\mathcal{CM}_j = \{cm_{j,1}, cm_{j,2}, \ldots, cm_{j,n_j}\}$), and each core has $g_j$ computing capacity. In edge $e_k$, $s_k$ servers are deployed, which are represented as $\mathcal{S}_k = \{s_{k,1}, s_{k,2}, \ldots, s_{k,s_k}\}$. Each edge server (say $s_{k,l}$) has $n_{k,l}$ cores, each with $g_{k,l}$ computing capacity. For satisfying users' requirements when local and edge resources are not enough, assuming there are $V$ cloud servers[4] ($\mathcal{V} = \{v_1, v_2, \ldots, v_V\}$) rented from the cloud. Cloud server $v_r$ has $n_{v_r}$ cores, and the capacity of each core is $g_{v_r}$. The price of $v_r$ is $p_r$ per unit time. The bandwidth of transmitting data from device $m_j$ to $s_{k,l}$ and $v_r$ are represented as $b_{j,k,l}$ and $b_{j,r}$, respectively, which can be

[3]Multiple clouds can be seen as one big public cloud including the resources provisioned by these clouds.

[4]The cloud resources can be provided in the form of virtual machine (VM), physical machine (PM), or both. The form of resource provisioning does not affect the application of our method.

easily calculated by transmission channel state data (*Chen et al., 2016*; *You et al., 2017*; *Du et al., 2019*). If a device is not in the coverage of an edge, corresponding bandwidths are set to infinity.

Assuming there are $T$ tasks, $\mathcal{T} = \{t_1, t_2, \ldots, t_T\}$, requested by users for processing in the DE3C, and each task can be processed in the device launching it (locally), an edge server communicated with the device, or a cloud server. For each task, say $t_i$, it has $in_i$ data must be processed, and has $f_i$ processing length. To make our method wider application, assuming the data size to be processed and the processing length are independent for each task in any computing node. Task $t_i$ must be finished before its deadline $d_i$ defined by corresponding SLA.[5] Assuming $d_1 \leq d_2 \leq \ldots \leq d_T$ without loss of generality. The following binary variables are defined for the following formulations.

[5]This paper considers hard deadline, where the service provider has no gains for a task if the task is finished after its deadline. The scheduling of soft deadline tasks is considered as a future work.

$$x_{i,j} = \begin{cases} 1, & \text{if } t_i \text{ is launched by } m_j \\ 0, & \text{otherwise} \end{cases}, \forall i \in [1, T], \forall j \in [1, M]. \tag{1}$$

$$x_{i,j,h} = \begin{cases} 1, & \text{if } t_i \text{ is processed by hth core in } m_j \\ 0, & \text{otherwise} \end{cases}, \forall h \in [1, n_j], \forall i \in [1, T], \forall j \in [1, M]. \tag{2}$$

$$y_{i,k,l,h} = \begin{cases} 1, & \text{if } t_i \text{ is processed by hth core in } s_{k,l} \\ 0, & \text{otherwise} \end{cases}, \forall h \in [1, n_{k,l}], \forall i \in [1, T], \forall j \in [1, M],$$
$$\forall l \in [1, s_k]. \tag{3}$$

$$z_{i,r,h} = \begin{cases} 1, & \text{if } t_i \text{ is processed by hth core in } v_r \\ 0, & \text{otherwise} \end{cases}, \forall h \in [1, n_{v_r}], \forall i \in [1, T], \forall r \in [1, V]. \tag{4}$$

This paper does not consider the cooperative computing between devices (*Hong et al., 2019*), and thus each task cannot be processed by other users' devices, *i.e.*,

[6]In this paper, the redundant execution for a task is not employed to improve the task performance (*Liu et al., 2019a*) as it cost more resources. Each task is assumed to be processed by only one core, as done in many published articles. A task executed on more than one cores usually can be decomposed into several subtasks. One is suggested to refer to our previous work (*Wang et al., 2019a*) which studied on the task scheduling with parallelism awareness, which complements this work.

$$x_{i,j,h} = 0, \forall h \in [1, n_j], \forall x_{i,j} = 0, \forall i \in [1, T], \forall j \in [1, M]. \tag{5}$$

Each task can be only assigned to one core in the DE3C,[6] and thus

$$\sum_{j=1}^{M} \sum_{h=1}^{n_j} x_{i,j,h} + \sum_{k=1}^{E} \sum_{l=1}^{s_k} \sum_{h=1}^{n_{k,l}} y_{i,k,l,h} + \sum_{r=1}^{V} \sum_{h=1}^{n_{v_r}} z_{i,r,h} \leq 1, \forall i \in [1, T]. \tag{6}$$

Then the number of completed tasks is

$$N = \sum_{i=1}^{T} \left( \sum_{j=1}^{M} \sum_{h=1}^{n_j} x_{i,j,h} + \sum_{k=1}^{E} \sum_{l=1}^{s_k} \sum_{h=1}^{n_{k,l}} y_{i,k,l,h} + \sum_{r=1}^{V} \sum_{h=1}^{n_{v_r}} z_{i,r,h} \right). \tag{7}$$

when $t_i$ is processed locally, its execution time is its computing time as there is no data transmission, *i.e.*,

$$\tau_i = f_i / g_j, \forall x_{i,j} = 1, \forall i \in [1, T], \forall j \in [1, M]. \tag{8}$$

As all tasks assigned to a core can be finished before their respective deadlines if and only if each task can be finished within its deadline when they are processed in ascending order of the deadline (*Pinedo, 2016*), the finish time of each task processed by a core can be calculated by assuming tasks are processed in the EDF scheme in the core. Then, if $t_i$ is processed in the $h$th core of device $m_j$, i.e., $x_{i,j,h} = 1$, its start time is the accumulated execution time of tasks that have earlier deadlines than it and are processed in the same core, i.e., $\sum_{ii=1}^{i-1}(x_{ii,j,h} \cdot \tau_{ii})$. Thus, the finish time of $t_i$ can be formulated as

$$ft_{i,j,h} = x_{i,j,h} \cdot \sum_{ii=1}^{i}(x_{ii,j,h} \cdot \tau_{ii}), \forall h \in [1, n_j], \forall i \in [1, T], \forall j \in [1, M]. \tag{9}$$

where $ft_{i,j,h} = 0$ when $t_i$ is not assigned to $h^{th}$ core of device $m_j$. In this situation, the deadline constraints can be formulated into

$$ft_{i,j,h} \leq d_i, \forall h \in [1, n_j], \forall i \in [1, T], \forall j \in [1, M]. \tag{10}$$

when a task is offloaded to an edge or the cloud, its execution time is formed from the data transfer time and computing time. If a task is scheduled to a core of an edge server or a cloud server, its computing can be started only when its data transmission is complete and the core has finished all tasks that have earlier deadlines and are assigned to the core (recall that EDF scheduling provides the best solution for a core in SLA optimization). The earliest completion time of data transmission for a task offloaded to an edge server or the cloud is

$$dt_{i,k,l,h} = y_{i,k,l,h} \cdot \sum_{ii=1}^{i}(y_{ii,k,l,h} \cdot \frac{in_{ii}}{b_{j,k,l}}), \forall h \in [1, n_{k,l}], \forall i \in [1, T], \forall j \in [1, M], \forall l \in [1, s_k], \tag{11}$$

or

$$dt_{i,v_r,h} = z_{i,r,h} \cdot \sum_{ii=1}^{i}(z_{ii,r,h} \cdot \frac{in_{ii}}{b_r}), \forall h \in [1, n_{v_r}], \forall i \in [1, T], \forall r \in [1, V]. \tag{12}$$

And the ready time of a core for computing an offloaded task in an edge or the cloud is the latest finish time of tasks that are offloaded to the same core and have earlier deadlines, which can be formulated into

[7] Usually the amount of result data is very small, and its transmission time is negligible compared with that of input data or the computing time. Thus, in this paper, the transmission time of the result data is ignored, as done by many previous works (*Zhao & Zhou, 2019*; *Chen & Hao, 2018*; *Hong et al., 2019*).

$$rt_{i,k,l,h} = y_{i,k,l,h} \cdot \max_{ii=1}^{i-1}(y_{ii,k,l,h} \cdot ft_{ii,k,l,h}), \forall h \in [1, n_{k,l}], \forall i \in [1, T], \forall j \in [1, M],$$
$$\forall l \in [1, s_k], \tag{13}$$

or

$$rt_{i,v_r,h} = z_{i,r,h} \cdot \sum_{ii=1}^{i-1}(z_{ii,r,h} \cdot ft_{ii,v_r,h}), \forall h \in [1, n_{v_r}], \forall i \in [1, T], \forall r \in [1, V], \tag{14}$$

where $ft_{i,k,l,h}$ and $ft_{i,v_r,h}$ represent the finish time of task $t_i$ when it is assigned to $h$th core in the edge server $s_{k,l}$ and the cloud server $v_r$ respectively. For a core processing an offloaded task, its start time is a later of its ready time and the completion time of the input data.[7] Thus, the finish time of offloaded tasks can be calculated as followings

$$ft_{i,k,l,h} = y_{i,k,l,h} \cdot (\max\{dt_{i,k,l,h}, rt_{i,k,l,h}\} + \frac{f_i}{g_{k,l}}), \forall h \in [1, n_{k,l}], i \in [1, T],$$

$$\forall l \in [1, s_k], \forall k \in [1, E]. \tag{15}$$

$$ft_{i,v_r,h} = z_{i,r,h} \cdot (\max\{dt_{i,v_r,h}, rt_{i,v_r,h}\} + \frac{f_i}{g_{v_r}}), \forall h \in [1, n_{v_r}], \forall i \in [1, T], \forall r \in [1, V]. \tag{16}$$

And the deadline constraints of task processing in edges and the cloud can be formulated into

$$ft_{i,k,l,h} \leq d_i, \forall h \in [1, n_{k,l}], i \in [1, T], \forall j \in [1, M], \forall l \in [1, s_k]. \tag{17}$$

$$ft_{i,v_r,h} \leq d_i, \forall h \in [1, n_{v_r}], \forall i \in [1, T], \forall r \in [1, V]. \tag{18}$$

For a computing node (a device, an edge server or a cloud server), the time occupied for processing tasks is the latest finish time of all tasks assigned to the node, *i.e.*,

$$ut_{m_j} = \max_{i=1}^{T} \max_{h=1}^{n_j} ft_{i,j,h}, \forall j \in [1, M], \tag{19}$$

$$ut_{s_{k,l}} = \max_{i=1}^{T} \max_{h=1}^{n_{k,l}} ft_{i,k,l,h}, \forall l \in [1, s_k], \forall k \in [1, E], \tag{20}$$

$$ut_{v_r} = \max_{i=1}^{T} \max_{h=1}^{n_{v_r}} ft_{i,v_r,h}, \forall r \in [1, V], \tag{21}$$

where $ut_{\mathcal{N}}$ is the use time of the computing node $\mathcal{N}$ for finishing tasks assigned to it. Then, the occupied resource amount ($or_{\mathcal{N}}$) of a computing node for task processing are respectively

$$or_{m_j} = ut_{m_j} \cdot g_j \cdot n_j, \forall j \in [1, M], \tag{22}$$

$$or_{s_{k,l}} = ut_{s_{k,l}} \cdot g_{k,l} \cdot n_{k,l}, \forall l \in [1, s_k], \forall k \in [1, E], \tag{23}$$

$$or_{v_r} = ut_{v_r} \cdot g_{v_r} \cdot n_{v_r}, \forall r \in [1, V], \tag{24}$$

And the consumed computing resource amount $cr_{\mathcal{N}}$ can be quantified by the accumulated processing length of its finished tasks for each computing node, *i.e.*,

$$cr_{m_j} = \sum_{i=1}^{T} \sum_{h=1}^{n_j} x_{i,j,h} \cdot f_i, \forall j \in [1, M], \tag{25}$$

$$cr_{s_{k,l}} = \sum_{i=1}^{T} \sum_{h=1}^{n_{k,l}} y_{i,k,l,h} \cdot f_i, \forall l \in [1, s_k], \forall k \in [1, E], \tag{26}$$

$$cr_{v_r} = \sum_{i=1}^{T} \sum_{h=1}^{n_{v_r}} z_{i,r,h} \cdot f_i, \forall r \in [1, V], \tag{27}$$

Thus, the computing resource utilization of devices, edge servers, and cloud servers are respectively

$$U_{device} = \frac{\sum_{j=1}^{M} cr_{m_j}}{\sum_{j=1}^{M} or_{m_j}}, \tag{28}$$

$$U_{edge} = \frac{\sum_{k=1}^{E} \sum_{l=1}^{s_k} cr_{s_{k,l}}}{\sum_{k=1}^{E} \sum_{l=1}^{s_k} or_{s_{k,l}}}, \tag{29}$$

$$U_{cloud} = \frac{\sum_{r=1}^{V} cr_{v_r}}{\sum_{r=1}^{V} or_{v_r}}, \tag{30}$$

and the overall resource utilization of the DE3C system is

$$U = \frac{\sum_{j=1}^{M} cr_{m_j} + \sum_{k=1}^{E} \sum_{l=1}^{s_k} cr_{s_{k,l}} + \sum_{r=1}^{V} cr_{v_r}}{\sum_{j=1}^{M} or_{m_j} + \sum_{k=1}^{E} \sum_{l=1}^{s_k} or_{s_{k,l}} + \sum_{r=1}^{V} or_{v_r}}. \tag{31}$$

Based on the above formulation, the task scheduling problem optimizing the SLA satisfaction can be modelled as

**Maximizing** $N + U$ $\tag{32}$

subject to

$(1) - (31)$ $\tag{33}$

where the objective Eq. (32) is to maximize the number of finished tasks, and to maximize the overall computing resource utilization when the finished task number cannot be improved. The decision variables include $x_{i,j,h}$ ($h \in [1, n_j], j \in [1, M], i \in [1, T]$), $y_{i,k,l,h}$ ($h \in [1, n_{k,l}], l \in [1, s_k], k \in [1, E], i \in [1, T]$), and $z_{i,r,h}$ ($h \in [1, n_{v_r}], r \in [1, V], i \in [1, T]$). This problem is binary nonlinear programming (BNLP), which can be solved by existed tools, *e.g.*, lp_solve (*Berkelaar et al., 2020*). These tools are implemented based on branch and bound, which is not applicable to large-scale problems. Therefore, a heuristic method is proposed to solve the problem in polynomial time in the next section.

## THREE-STAGE HEURISTIC TASK SCHEDULING

This section presents the proposed hybrid heuristic method, called TSSLA (Three-Stage scheduling optimizing SLA), to address the task scheduling problem stated in the previous section, which coordinates the richness of cloud computing resources and the low transmission delay of local and edge resources, to optimize the SLA satisfaction.

---

**Algorithm 1** The three-stage hybrid heuristic task scheduling

---

**Input:** the information of tasks and DE3C resources
**Output:** the mapping between tasks and computing resources
    //The first stage
 1: iteratively pre-assigning a task to a new cloud server core, until no task can be finished in the cloud;
 2: iteratively pre-assigning a task to an edge server core using Algorithm 2, until no task can be finished in edges;
 3: iteratively assigning a task to a device core using Algorithm 2, until no task can be finished locally;
    //The second stage
 4: for all tasks assigned to edges, iteratively re-assigning a task to a device core using Algorithm 2, until no task can be finished locally;
 5: for remaining unassigned tasks, iteratively assigning a task to an edge server core using Algorithm 2, until no task can be finished in edges;
    //The third stage
 6: for all tasks assigned to the cloud, iteratively re-assigning a task to a device core using Algorithm 2, until no task can be finished locally;
 7: for all tasks assigned to the cloud, iteratively re-assigning a task to an edge server core using Algorithm 2, until no task can be finished locally;
 8: for tasks assigned to the cloud, iteratively re-assigning the task in the next cloud VM to one of previous VMs using Algorithm 2, and de-renting idle cloud VMs;

---

The proposed hybrid heuristic method includes three stages, where the first stage tries to satisfy deadlines of as many tasks as possible, by prioritising the usage of abundant cloud computing resources and assigning tasks that cannot be finished in the cloud to edges or corresponding devices. In the second stage, TSSLA tries to make full use of local resources and release some edge resources by rescheduling several tasks from edges to corresponding devices, and to exploit edge resources shared by multiple users for finishing more tasks. At the last stage, our method aims at optimizing the cost of cloud resources, by rescheduling as many tasks as possible from the cloud to corresponding devices and edges. Algorithm 1 outlines the proposed three-stage hybrid heuristic task scheduling.

As shown in Algorithm 1, in the first stage, TSSLA first pre-assigns all tasks that can be finished by cloud resources to the cloud (line 1 in Algorithm 1). For each task, TSSLA pre-rents a one-core VM instance with the best cost performance. In real world, a public cloud, *e.g.*, Amazon EC2 (*Amazon, 2020*), provides various VM types configured with different core numbers, and for each type[8], *e.g.*, c6g.* in Amazon EC2, VM instances have a same price per core. Thus, TSSLA pre-rents one-core VM instances in this stage. After this step, there is no task can be finished in the cloud, and TSSLA pre-assigns remain tasks to edge servers employing LASTF and EDF for computing core selection and task ordering in each core, respectively (see Algorithm 2). Then, TSSLA schedules tasks to each device adopting Algorithm 2.

At the second stage, TSSLA examines each task assigned to edges. If the task can be finished in its device, TSSLA reassigns the task to the device, and the edge has more available resources for processing unassigned tasks. Thus, after that, TSSLA repeats step 2 in the first stage, which assigns remain tasks to edge servers by Algorithm 2. Now, no more tasks can be finished by DE3C resources, thus TSSLA only improves the resource usage in the last stage.

TSSLA employs two approaches to improve the resource efficiency in the third stage. One approach is trying to reassign as many tasks as possible from the cloud to local devices and edge servers because local resources and edge server resources are cheaper and have much less network latency, compared with cloud resources. And another is to consolidate

---

[8]This paper assumes that cloud resources are provided on demand, and leave the concern of other provisioning scheme, *e.g.*, spot, as a future work.

---

**Algorithm 2** Scheduling a task in a device, edges or the cloud

---

**Input:** the information of the task ($t$), computing cores and tasks assigned to each core
**Output:** the computing core the task is assigned to, and **False** if the task cannot be finished by any core
1:  $LAST \leftarrow +\infty$;//recording the least accumulated slack time
2:  **for** each core **do**
3:      assuming $t$ is assigned to the core, calculating the finish time of each task
        assigned to the core, using (8) and (9) for device cores, (11), (13) and (15)
        for edge server cores, and (12), (14) and (16) for cloud VMs;
4:      **if** the finish time of one task is later than its deadline **then**
5:          **continue**;
6:      **end if**
7:      calculating the slack time of each task, which is the difference between its finish
        time and its deadline;
8:      accumulating each slack time of all tasks;
9:      **if** the accumulated slack time $< LAST$ **then**
10:         $LAST \leftarrow$ the accumulated slack time;
11:         recording the core;
12:     **end if**
13: **end for**
14: **if** $LAST < +\infty$ **then**
15:     **return** the last recorded core;
16: **else**
17:     **return False**;
18: **end if**

---

tasks assigned to the cloud for improving the cost efficiency by reducing the idle time of VM instances, as cloud resources are charged by time unit, *e.g.*, hour.

Therefore, in the third stage, TSSLA examines each task assigned to the cloud, and if the task can be finished by corresponding device, TSSLA reassigns the task to the device (see line 6 in Algorithm 1). Otherwise, TSSLA checks whether the task's requirements can be satisfied by an edge server, and if so reassigns the task to the edge server (see line 7 in Algorithm 1). After these reassignments, TSSLA reassigns the task that has assigned to the next VM to one of its previous VMs by using Algorithm 2, and re-rents idle VMs. This step reduces the idle time of VMs, which improves the cost efficiency, as the rent time of each VM is round up to times of the charge unit for its cost. For example, if one user rents a VM for 1.8 h, and the cloud provider charges \$0.1 per hours, the user must pay \$0.2 (\$0.1/hours $\times \lceil 1.8 \rceil$ hours) for the VM.

TSSLA employs Algorithm 2 to implement all of these above task assignments in each device, edges, and the cloud, which decides which core to process the task. The detail is shown in the following.

As shown in Algorithm 2, to select an available computing core for a task, TSSLA traverses each available core (line 2 in Algorithm 2), and calculates the accumulated slack time with the assumptions that the task is assigned to the core and all tasks are executed in the ascending order of their deadlines (lines 3–8 in Algorithm 2). Then, TSSLA allocates the core providing LAST to the task (lines 9–15 in Algorithm 2). For each available core, if the assignment of the task results in at least one deadline violation (line 4 in Algorithm 2), Algorithm 2 returns false which means the requirements of the task cannot be satisfied by any of these available computing cores (line 17 in Algorithm 2).

# RESULTS

This section conducts simulated experiments designed by referring to related works and real worlds, to evaluate the performance of the proposed method.

**Table 2  The parameters of simulated DE3C system.**

| Tasks | | | Device | Edge server | Cloud VM |
|---|---|---|---|---|---|
| Number | 100 | Number | 10 | [1, 4] | – |
| Computing length | [1, 2000] GHz | Core number | [1, 4] | [4, 8] | – |
| Processed data size | [20, 500] MB | Capacity per core | [1, 2] GHz | [2, 3] GHz | [2, 3] GHz |
| Deadline | [500, 1500] s | Bandwidth | – | [10, 100] Mbps | [1, 10] Mbps |
| | | Price | – | – | $0.01/hour |

## Experiment design

A DE3C system is established, which is composed of one public cloud, an edge, and 10 user devices. The set of various system parameters are referring to that of *Du et al. (2019)*, *Chen et al. (2016)*, *Alkhalaileh et al. (2020)*, the Gaia Cluster (*University of Luxembourg, 2020*), as well as Amazon EC2 (*Amazon, 2020*), which are detailed as followings and shown in Table 2. One hundred tasks are generated randomly. Each task is randomly associated to one device that is regarded as the device launching the task. The length and the size of each task are randomly set in ranges of [1, 2000] GHz and [20, 500] MB, respectively, to cover small to large tasks. The computing capacities of each core in a device, an edge server, a cloud VM are randomly in the ranges [1, 2] GHz, [2, 3] GHz, and [2, 3] GHz, respectively. The number of computing core is randomly set in the ranges [1, 4] and [4, 8], respectively for each device and each edge server. The number of servers is set in [1, 4] for the edge. The price of each core is $0.01 per hour for cloud VMs. The bandwidths for transmitting data from a device to the edge and the cloud are in [10, 100] Mbps and [1, 10] Mbps respectively.

The performance of TSSLA is compared with the following classical or state-of-the-art scheduling methods designed for DE3C system. As done in all of the existed works (the best of our knowledge), the following methods employ local resources first, and the cloud resource at last, for processing tasks.

- **FF** (First Fit) (*Bays, 1977*) iteratively assigns the first task to the first core satisfying its requirements.
- **FFD** (First Fit Decreasing) (*B.V. & Guddeti, 2018*) iteratively assigns the largest task to the first core satisfying its requirements.
- **EDF** (Earliest Deadline First) (*Benoit, Elghazi & Robert, 2021*) iteratively assigns the task with the earliest deadline to the first core satisfying its requirements.
- **BF** (Best Fit) (*Zhao & Kim, 2020*) iteratively assigns the first task to the core that satisfies its requirements and provides the latest finish time for the task.
- **EFTF** (Earliest Finish Time First) which is the basic idea of the method proposed by *Liu et al. (2019a)*, iteratively assigns the first task to the core providing the earliest finish time.
- **EDF_EFTF**, the idea of Stavrinides's and Karatza's proposed method (*Stavrinides & Karatza, 2019*), respectively employs EDF and EFTF for task selection and resource selection.

- **LSTF** (Least Slack Time First) (*Michel et al., 2021*) iteratively assigns the first task to the core providing the least slack time.
- **LSSRF** (Least Size-Slack time ratio First), the idea employed by *Mahmud et al. (2020)*, iteratively assigns the first task to the core providing the maximal ratio between the profit and the slack time. The length is regarded as the profit for each task.

The performance metrics used to quantify the performance of each task scheduling method include the followings.

- **SLA satisfaction** can be quantified by the amount of finished tasks in number, length, and processed data size, which are respectively calculated by Eqs. (7), (34), and (35). The larger value is better for the metric. For the length and the processed data size of finished tasks, the followings report the percentages of that of all launched tasks.

$$len = \sum_{i=1}^{T} ((\sum_{j=1}^{M} \sum_{h=1}^{n_j} x_{i,j,h} + \sum_{k=1}^{E} \sum_{l=1}^{s_k} \sum_{h=1}^{n_{k,l}} y_{i,k,l,h} + \sum_{r=1}^{V} \sum_{h=1}^{n_{v_r}} z_{i,r,h}) \cdot f_i). \tag{34}$$

$$size = \sum_{i=1}^{T} ((\sum_{j=1}^{M} \sum_{h=1}^{n_j} x_{i,j,h} + \sum_{k=1}^{E} \sum_{l=1}^{s_k} \sum_{h=1}^{n_{k,l}} y_{i,k,l,h} + \sum_{r=1}^{V} \sum_{h=1}^{n_{v_r}} z_{i,r,h}) \cdot in_i). \tag{35}$$

- **Resource utilization** is one of the most popular metrics to quantify the resource efficiency, which is the ratio between amounts of consumed resources and occupied resources, *i.e.,* $U$ calculated by Eq. (31). It is better for a higher value.
- **Makespan** is the latest finish time of tasks, which can be achieved by Eq. (36). Earlier makespan means faster processing rate, and thus is better.

$$makespan = \max_{i=1}^{T} \{\max\{\max_{j=1}^{M} \max_{h=1}^{n_j} ft_{i,j,h}, \max_{k=1}^{E} \max_{l=1}^{s_k} \max_{h=1}^{n_{k,l}} ft_{i,k,l,h}, \max_{r=1}^{V} \max_{h=1}^{n_{v_r}} ft_{i,v_r,h}\}\}. \tag{36}$$

- **Cost efficiency** is the length of tasks processed by per-dollar resource in the cloud, as calculated by Eq. (37). It is a metric for quantifying the resource efficiency in clouds. A greater value is better.

$$Ceff = \frac{len}{\sum_{r=1}^{V} \lceil cr_{v_r} \cdot p_r \rceil} \tag{37}$$

## Experiment results
### SLA satisfaction

Figure 2 shows the performance of various task scheduling methods in SLA satisfaction. As shown in the figure, TSSLA has 22.2%–27.6%, 47.3%–59.1%, and 25.4%–32.6% better SLA satisfaction performance compared with other methods in task number, task computing length, and processed data size, respectively. The superiority of our method is allocating computing resources according to the scarcity degree of resources. TSSLA prefers using the abundant computing resources of the cloud, and employs scarce computing resources of edges and devices for processing tasks cannot be finished by the cloud due to the poor network performance between users and their cloud, in its first stage. In contrary, other

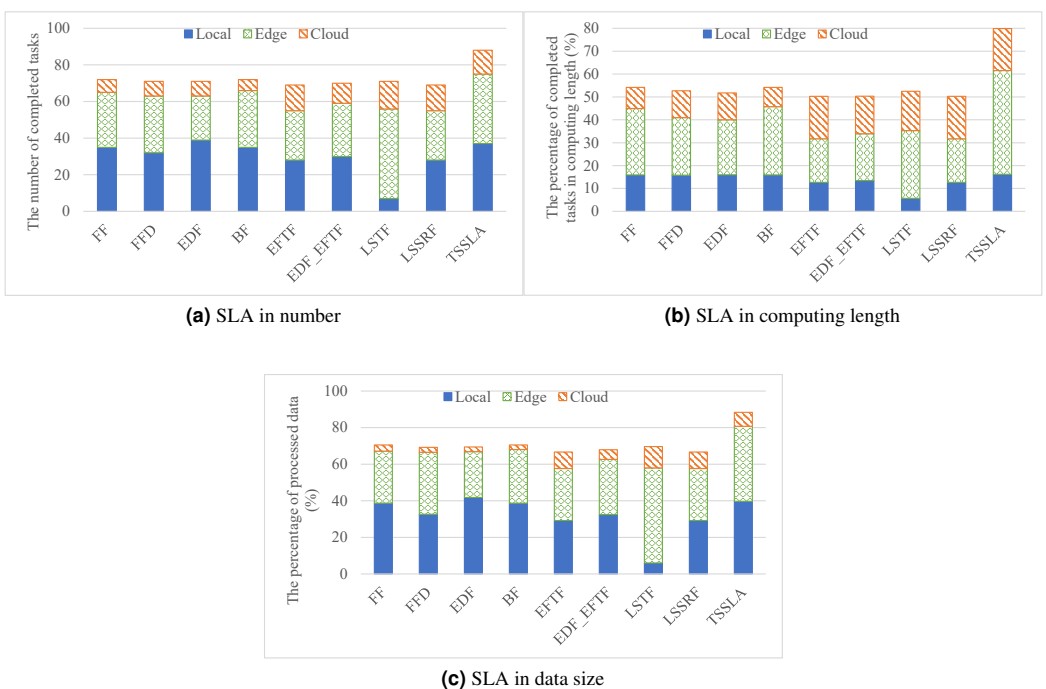

**(a)** SLA in number

**(b)** SLA in computing length

**(c)** SLA in data size

**Figure 2** The SLA satisfaction performance in task number, task computing length, and processed data size respectively, achieved by various scheduling methods.

methods prefer local resources or nearby edge resources, aiming at providing the best performance for each task with minimal resource costs. But these methods result in several local and nearby edge resources that are used by some tasks which can be finished by the cloud, and these resources can be reserved for processing other tasks whose demands cannot be satisfied by the cloud. Thus, our method has a better performance than other methods in SLA satisfaction. Based on the idea of our method, these works can be improved by reassigning some tasks from local devices or edges to the cloud, to make some resources idle for finishing remaining unassigned tasks.

Figure 2 also shows that, except TSSLA, EDF has the best performance in optimizing SLA satisfaction in devices, and LSTF achieves the most number of completed tasks in the edge. Even so, all methods except TSSLA have comparable performance in SLA optimization overall. The reason why EDF is better than FF, FFD, BF and EFTF in SLA optimization in the devices is that EDF prioritizes the demands of tasks with tight deadlines, and thus postpones tasks with more slack time, which can finish more tasks with tight deadlines compared with other methods. Besides, EDF yields an optimal schedule for maximizing the number of finished tasks in each core (*Pinedo, 2016*). After completing more tasks locally, there are fewer tasks can be finished in edges or the cloud, as shown in Fig. 2, when applying EDF. This phenomenon does not occur when employing TSSLA. TSSLA satisfies more tasks not only in local devices but also in the edge, compared with other methods (except EDF in devices and LSFT in the edge). This is mainly because TSSLA assigns tasks that can be finished by both the cloud and local devices or the edge to the cloud at the first, which

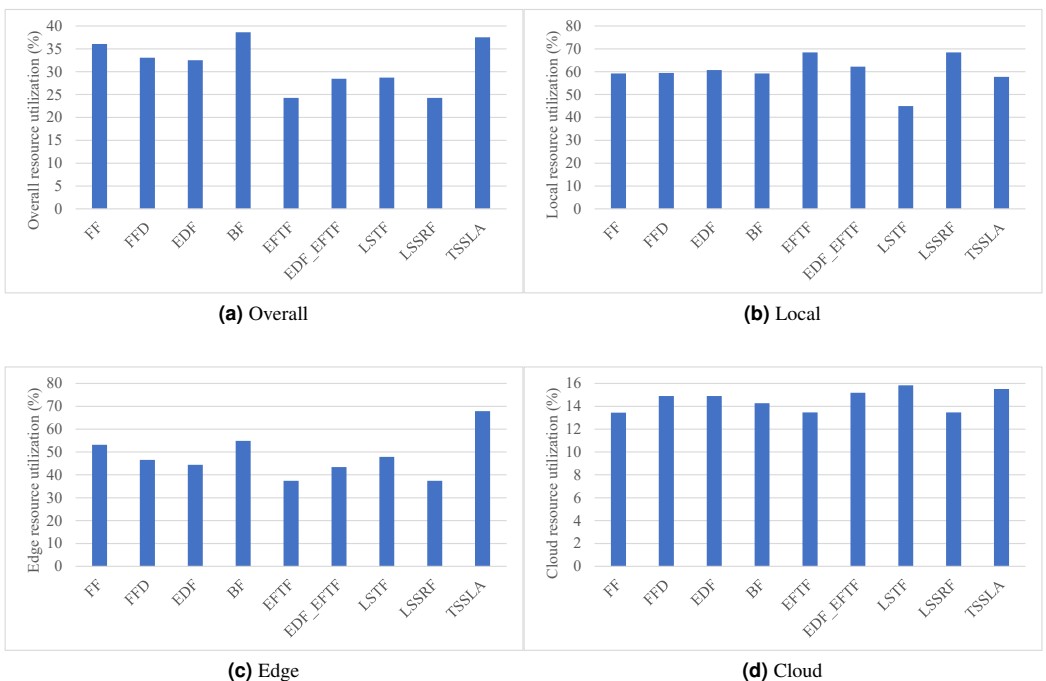

**Figure 3** (A–D) The overall computing resources of the DE3C, devices, edge servers, and cloud servers, when applying various scheduling methods.

results in more available local and edge resources for processing tasks whose demands can be satisfied only by local devices or the edge. This further verifies the high efficiency of our method. The main reason why LSTF provides the most number of completed tasks at the edge is that it completes much fewer tasks by local resources, compared with other methods, and thus leaves more tasks with loose deadlines to the edge for processing.

### Resource utilization

As shown in Fig. 3, TSSLA achieves almost same overall resource utilization to BF, and 4.1%–54.6% higher than other methods, which verifies that our method provides high resource efficiency for task processing in DE3C environments. The reason that TSSLA achieves a higher overall resource utilization, compared with other methods, is because it provides a much better utilization than others in the edge, as shown in Fig. 3C. In addition, TSSLA completes more than half of tasks' computing length at the edge, as shown in Fig. 2B. In general, more tasks processed by the edge or the cloud means higher resource utilization in the edge or the cloud (shown in Figs. 2, 3C, and 3D). This is because more tasks can result in less ratio between the amounts of idle computing resources and occupied computing resources, as the data transmission and the data computing can be parallel for different tasks, in each core. Compared with LSTF, TSSLA achieves 41.7% higher utilization, as shown in Fig. 3C, although it completes 22.5%, 13.3%, and 21.1% less tasks in number, computing length, and processed data size, respectively, as shown in Fig. 2, in the edge. The main reason is that TSSLA has a greater ratio between the computing

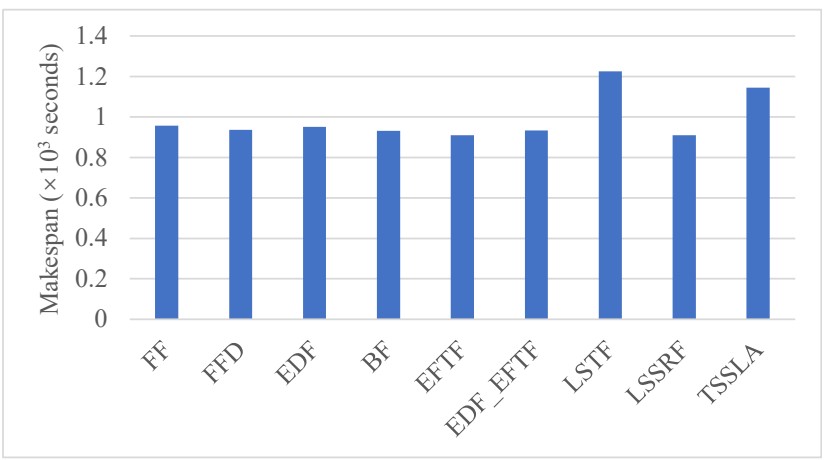

**Figure 4** The latest finish time of tasks in the DE3C, when applying various scheduling methods.

length and the processed data size of tasks completed by the edge than LSTF, which leads to less idle computing resources, and thus results in a higher computing resource utilization. This phenomena can be exploited to design heuristic scheduling methods for optimizing the efficiency of computing resources in edges and clouds.

The reason why BF achieves the highest overall resource utilization, as shown in Fig. 3A, is that the tasks processed by the cloud is the least in each SLA metric, as shown in Fig. 2, and thus the low utilization of cloud computing resources has the lowest impact on the overall resource utilization.

The utilization of cloud computing resources is much less than that of edge computing resources for each method, as shown in Fig. 3D, which is because the network performance of the cloud is much worse, leading more idle computing resources due to the longer time of data transmission, compared with that of the edge. Thus, it would be good to assign tasks with small data sizes to the cloud. Tasks processing less data usually have larger computing length, and thus there is a trade-off between the limited computing resources of devices or edges and the poor network performance of the cloud, which is one of our considerations for designing highly efficient scheduling methods in future.

### *Makespan*

Figure 4 shows the makespan when applying various scheduling methods. As shown in the figure, TSSLA has a larger makespan than other methods except LSTF, as the DE3C completes the most computing length and processes the most data of tasks when applying TSSLA, and the makespan is usually increased with the completed computing length and the processed data size. In fact, TSSLA has only about 20% larger makespan, but it completes more than 47.3% more computing length and processes more than 25.4% more data than other methods except LSTF. In addition, LSTF has larger makespan to TSSLA, even though it completes much less computing length and processes much fewer data than TSSLA. These results validate the efficiency of our methods further.

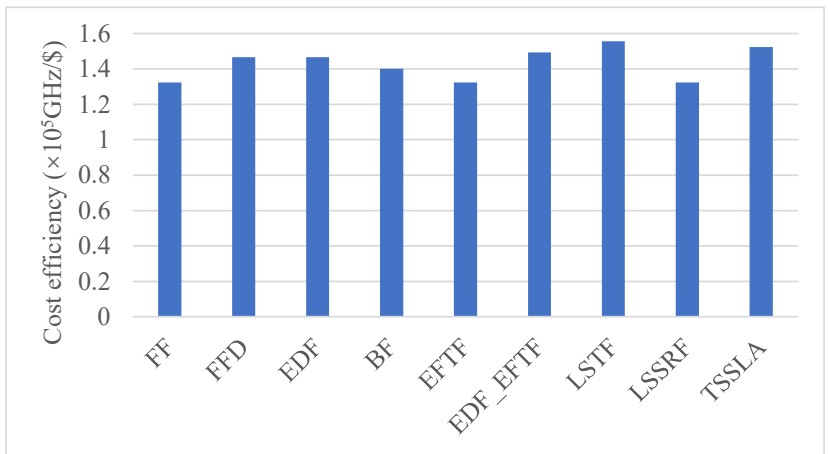

**Figure 5** **The cost efficiency of cloud computing resources, when applying various scheduling methods.**

The major reason of LSTF having the largest makespan is that it completes the most number of tasks in both the edge and the cloud, and the local device provides much less processing time than an edge and especially the cloud for a task due to the data transmission time from the device to the edge and the cloud.

### Cost efficiency

Figure 5 shows the cost efficiency for task processing in the cloud. TSSLA has a comparable cost efficiency to LSTF and is better than others. Comparing Figs. 3D and 5, we can see that the relative performance of the cost efficiency is almost same as that of the computing resource utilization. This is mainly because the cloud computing resources are charged by the use time. Thus, in most time, the resource utilization and the cost efficiency are equivalent for quantifying the usage efficiency of cloud computing resources. The reason why LSTF has a good cost efficiency is that it completes the most number of tasks in the cloud, as illustrated in 'Resource Utilization'.

## CONCLUSIONS

This paper studies on the SLA satisfaction optimization in a device-edge-cloud cooperative computing (DE3C) environment. This paper first formulates the problem into a BNLP, and then proposes a heuristic scheduling method, named TSSLA, to solve the problem in polynomial-time complexity. TSSLA consists of three heuristic stages which respectively exploit the abundant computing resources of the cloud, the shared resources of edges, and the low/zero network latency of edge and device resources, for optimizing the number of tasks whose requirements are satisfied and the resource efficiency. Experiment results confirm the superior performance of TSSLA in optimizing SLA satisfaction and resource efficiency.

In fact, our method improves the SLA satisfaction and the resource efficiency by improving the collaboration ability among devices, edges, and clouds to exploit all of their

benefits. This idea can be also applied to other hybrid computing systems, *e.g.*, multi-clouds, hybrid clouds, which is one of our future work.

This paper focuses on the task scheduling for DE3C environments, assuming the data is transmitted to the computing node only when the offloading decision is made for each task. Caching data in edge servers and especially the cloud in advance can improve the performance of task executions. Thus, the prediction of offloading decisions and the caching strategy in DE3C will be studied in the future. In addition, the design of cache-aware task scheduling methods will be concerned to improve the benefits of caching strategies.

### Funding
The research was supported by the Key Scientific and Technological Projects of Henan Province (Grant No. 202102210174, 212102210096, 202102210383, 212102210410, 202102210149, 212102210382, 212102210104, 212102210424, 212102210418), the Key Scientific Research Projects of Henan Higher School (Grant No. 20B520039, 21A520050), the National Natural Science Foundation of China (Grant No. 61872043, 61975187, 62072414), Qin Xin Talents Cultivation Program, Beijing Information Science and Technology University (No. QXTCP B201904), and the fund of the Beijing Key Laboratory of Internet Culture and Digital Dissemination Research (Grant No. ICDDXN004). The funders had no role in study design, data collection and analysis, decision to publish, or preparation of the manuscript.

### Grant Disclosures
The following grant information was disclosed by the authors:
The Key Scientific and Technological Projects of Henan Province: 202102210174, 212102210096, 202102210383, 212102210410, 202102210149, 212102210382, 212102210104, 212102210424, 212102210418.
The Key Scientific Research Projects of Henan Higher School: 20B520039, 21A520050.
The National Natural Science Foundation of China: 61872043, 61975187, 62072414.
Qin Xin Talents Cultivation Program.
Beijing Information Science and Technology University: QXTCP B201904.
The Beijing Key Laboratory of Internet Culture and Digital Dissemination Research: ICDDXN004.

### Competing Interests
Junqiang Cheng is employed by Europe-Aisa Hi-tech and Digital Technology Company Limited.

### Author Contributions
- Yongxuan Sang conceived and designed the experiments, performed the experiments, prepared figures and/or tables, and approved the final draft.
- Junqiang Cheng conceived and designed the experiments, analyzed the data, authored or reviewed drafts of the paper, and approved the final draft.

- Bo wang conceived and designed the experiments, performed the computation work, authored or reviewed drafts of the paper, and approved the final draft.
- Ming Chen analyzed the data, prepared figures and/or tables, and approved the final draft.

## Data Availability

The implementation of task scheduling methods in C is available in the Supplementary File.

## Supplemental Information

Supplemental information for this article can be found online at http://dx.doi.org/10.7717/peerj-cs.851#supplemental-information.

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
