# Peer review of "A three-stage heuristic task scheduling for optimizing the service level agreement satisfaction in device-edge-cloud cooperative computing"

_PeerJ Computer Science, doi:10.7717/peerj-cs.851_

## Round 0.1 · original submission · Major Revisions

The authors should prepare a major revision in light of the comments.

·

Basic reporting

No comment

Experimental design

No comment

Validity of the findings

No comment

Additional comments

I would suggest the authors to incorporate the following very minor changes in the manuscript

1. I would suggest you to add a "motivation" sub-section and "contributions" sub-section in the "Introduction" section.

2. In line 73, the authors should add the reference of the paper from where the EDF technique is cited.

3. Similarly, on lines 245, 246, 248, 250, and 256 references to FF, FFD, EDF, BF, LSTF are missing, cite the papers from where these techniques are used in the current paper.

Reviewer 2 ·

Basic reporting

Authors present three-stage heuristic task scheduling for optimizing the service level agreement satisfaction in device-edge-cloud cooperative computing in this paper. Article is well written, and authors have done great job. However, there are some very serious question and comments which need to be addressed.
• Abstract: According to authors, they proposed the technique to solve the problem in reasonable time. What will be considered reasonable time?
• THREE-STAGE HEURISTIC TASK SCHEDULING: In the proposed technique, priority order is (i) clouds, (ii) edge servers, and (iii) local devices. Shouldn’t it be other way around? If they task can be executed on the device, it should run on the device. If it can’t, it should be placed to edge servers to reduce latency and overall cost (which will be paid to cloud service provider) and in the end if task cannot be placed on edge servers, due to limited resources, it should be forwarded to cloud. This mechanism will improve overall resource utilization of user device and edge server. Moreover, it will remove latency, cloud placement cost, and migration and rescheduling costs.
• THREE-STAGE HEURISTIC TASK SCHEDULING: In 2nd stage, tasks are shifted to device which can be accommodated on the device. Why its not done in first place?
• THREE-STAGE HEURISTIC TASK SCHEDULING: According to authors, last stage improves the resource cost. How will it improve resource cost? what about cost of such frequent migrations? why a smart technique is not devised which places the workload intelligently and reduces the cost of rescheduling and migration?

Experimental design

• Experimental Design, please provides the details/specifications of tasks, VMs and physical machines in tabular form.
• Experimental Design, please provide the proper reference number of all the selected techniques. How are these selected existing techniques closely related to your work? All of them are based on bin packing techniques. Why other state of the art, heuristic-based techniques are not considered for the comparison? Some of these selected techniques are very old and outdated.
• Experimental Design, please provide the formula of each selected performance metric.
• Results: According to authors, they achieved 59% better SLA satisfaction. How this satisfaction is achieved? And have they considered performance / SLA degradation due to migration? If yes, what is its impact? if no, why didn't you considered such an important consideration?

Validity of the findings

• Results: According to authors, they achieved 59% better SLA satisfaction. How this satisfaction is achieved? And have they considered performance / SLA degradation due to migration? If yes, what is its impact? if no, why didn't you considered such an important consideration?
• Results: Figures 2, 3, 4 and 5. What does y axis show in these graphs? Does it represent the graph heading or caption? Results are confusing. Values of y axis are in percentage or numbers? Please provide the proper axis titles in each graph along with the units.
• Results: SLA Satisfaction, “In contrary, other methods prefer local resources or nearby edge resources, aiming at providing the best performance for each task with minimal resource costs. But these methods result in several local and nearby edge resources that are used by some tasks which can be finished by the cloud, and these resources can be reserved for processing other tasks whose demands cannot be satisfied by the cloud.”, there are techniques that place the tasks on local device or edge, but when the real-time tasks with latency limitation is to be offloaded, they are placed (and given priority) on the local device/edge and non-real time tasks hosted on those machines are moved to next layer of hierarchy.
• Results: Resource utilization of existing techniques shouldn’t be high? They are utilizing local and edge resources first compared to cloud.
• Results: Cost Efficiency of proposed technique should be much higher compared to existing techniques as it first places the tasks on cloud rather than utilizing local resources.
• Results: Number of migration, performance degradation due to migrations and cost of migrations should be considered in performance evaluation process.
• Results: Rationale behind the result discussion is very weak.

Additional comments

• Related Work: limited related work. should be extended by including 2021 techniques.

·

Basic reporting

Problem that is tackled in proposed paper is interesting and important, however proposed manuscript suffers from some drawbacks, which are listed below.

According to my opinion proposed manuscript should undergo major revision before acceptance.

Experimental design

Visualization of results should be improved - consider using box and whiskers diagrams, swarm plots, etc. Also, please include convergence speed graphs, since the convergence is very important indicator of system performance.

Validity of the findings

Proposed method should be compared with SOTA (state-of-the-art) metaheuristics methods. Please, examine literature and include few metaheuristics approaches.

Moreover, statistical tests should be executed to prove significant results improvements over other methods.

Additional comments

Abstract should be completely reformulated to highlight main ideas and contributions of the proposed research. Abstract should emphasize a problem that is being solved, importance of the problem, employed methods and achieved results along with methods used in comparative analysis.

Your abstract does not highlight the specifics of your research or findings but contains too much background information. Some details of your research would be nice. An abstract with some details helps show the impact of your research. To aid in this, here is one of many good articles concerning crafting an abstract https://writing.wisc.edu/handbook/assignments/writing-an-abstract-for-your-research-paper.

It is not the best practice to use abbreviations in the abstract. Moreover, method names should not be capitalized.

Writing paper in the 1st person does not sound scientifically. Please rephrase all sentences into the 3rd person.

Introduction should be clearly presented to highlight main ideas and motivation behind the proposed research. Please include and clearly state research question and contributions of proposed study in Introduction.

Literature review should be improved to include metaheuristics-based approaches. There are many literature sources where the cloud-edge task scheduling problem was solved by using metaheuristics. You may consider the following reference:

https://www.mdpi.com/574168

Related works section should be included after Introduction, alternatively as a subsection within Introduction.

---

## Round 0.2 · accepted · Accept

The reviews indicate a significant improvement of the article.

·

Basic reporting

No comment

Experimental design

No comment

Validity of the findings

No comment

Additional comments

I am satisfied with the manuscript now as the asked changes have been incorporated by the authors. Hence this manuscript may be accepted for publication.

·

Basic reporting

Dear Authors,

thank you for addressing my comments.

However, I still noticed some minor English language and technical issues, therefore please read once again your manuscript thoroughly before publication.

All the best

Experimental design

no comment

Validity of the findings

no comment

Additional comments

Dear Authors,

thank you for addressing my comments.

However, I still noticed some minor English language and technical issues, therefore please read once again your manuscript thoroughly before publication.

All the best